# Low-Grade Endometrial Stromal Sarcoma: A Case Report of a Rare Uterine Malignancy Mimicking Degenerative Uterine Leiomyoma in a Nulliparous Woman

**DOI:** 10.3390/diagnostics15010018

**Published:** 2024-12-25

**Authors:** Hyun Kyung Lee, Weon Jang, Kyoung Min Kim, Ji Soo Song

**Affiliations:** 1Department of Radiology, Jeonbuk National University Medical School and Hospital, 20 Geonji-ro, Deokjin-gu, Jeonju 54907, Jeonbuk, Republic of Korea; radiologistlee@gmail.com (H.K.L.); weon0315@gmail.com (W.J.); 2Research Institute of Clinical Medicine, Jeonbuk National University, Jeonju 54907, Jeonbuk, Republic of Korea; kmkim84@jbnu.ac.kr; 3Biomedical Research Institute, Jeonbuk National University Hospital, Jeonju 54907, Jeonbuk, Republic of Korea; 4Department of Pathology, Jeonbuk National University Medical School and Hospital, 20 Geonji-ro, Deokjin-gu, Jeonju 54907, Jeonbuk, Republic of Korea

**Keywords:** endometrial stromal sarcoma, leiomyoma, magnetic resonance imaging, diffusion-weighted image, apparent diffusion coefficient

## Abstract

**Background and Clinical Significance:** Low-grade endometrial stromal sarcoma (LGESS) is a rare uterine malignancy that causes non-specific symptoms which presents more typically in younger women compared to other uterine sarcomas. Preoperative diagnosis of myometrial LGESS is challenging, as it is frequently mistaken for a benign uterine mass, such as a degenerating leiomyoma. Despite its rarity, the imaging findings of LGESS are highly variable, complicating the diagnostic process. Characteristic findings on magnetic resonance imaging T2-weighted imaging (T2WI)—including intra-tumoral low signal intensity (SI) bands (correlating with preserved myometrial bundles separated by tumor cells on histopathology), cystic/necrotic changes, and absence of a speckled appearance—have been significantly associated with LGESS. Additionally, apparent diffusion coefficient mapping can aid in the characterization of uterine masses. **Case Presentation**: We present a case of LGESS initially misdiagnosed as red degeneration of a uterine leiomyoma (RDL) due to a peripheral rim showing high SI on T1-weighted imaging and low SI on T2WI, which was interpreted as a thrombosed vessel. Histopathology demonstrated necrotic tissue outlined by normal uterine tissue, corresponding to the peripheral rim. We suggest that susceptibility-weighted imaging could have aided in distinguishing between the two conditions due to its high sensitivity to blood products. Moreover, diffusion-weighted imaging revealed restriction along T2 low SI bands, with no restrictions within the bands themselves, potentially indicating a viable tumor along preserved myometrium. **Conclusions**: These imaging features may provide valuable insights for diagnosing LGESS and differentiating it from RDL, supporting further research on LGESS imaging characteristics.

## 1. Introduction

Endometrial stromal sarcoma (ESS) is a rare uterine mesenchymal tumor composed of cells that histologically resemble endometrial stromal cells. It accounts for 7–25% of uterine sarcomas and is the second most common type after leiomyosarcoma, although it represents only 0.2% of all uterine malignancies [1,2]. Endometrial stromal sarcoma mostly occurs in the endometrium or myometrium of the uterus and occasionally in extrauterine locations, such as the ovaries, peritoneum, and vagina [3]. ESS can be classified into three major categories: low-grade ESS (LGESS), high-grade ESS, and undifferentiated uterine sarcoma.

LGESS, a less-aggressive subtype of ESS, more typically occurs in younger women during the premenopausal and perimenopausal phases compared to other subtypes of uterine sarcomas [4]. Although LGESS usually follows a slow clinical course, it can relapse and may lead to mortality [1,2]. The clinical presentation includes abnormal uterine bleeding and pelvic pain, which are non-specific and resemble the signs of other uterine lesions. Despite its generally indolent course and a five-year overall survival rate of 80–100%, the disease demonstrates malignant behavior, with 37–60% of patients experiencing late relapse and 15–20% eventually succumbing to the disease [1,2,4].

Although certain surgical approaches aim at preserving fertility, particularly in younger women [5], total abdominal hysterectomy without morcellation and bilateral salpingo-oophorectomy remain the recommended first-line treatments for LGESS [4]. Therefore, preoperative suspicion for uterine malignancy is important. However, due to its rarity, non-specific symptoms, and greater incidence among younger women, LGESS is frequently misidentified as other benign lesions such as leiomyoma or adenomyosis [6].

This diagnostic challenge was illustrated in a retrospective study of 153 cases of LGESS, where the preoperative diagnoses included leiomyoma in 109 cases (71.2%), low- or high-grade ESS in 19 cases (12.4%), adenomyosis in 9 cases (5.9%), and other diagnoses in 16 cases (10.5%) [7]. Additionally, a study by Chen et al. demonstrated that the most common unexpected malignancy after laparoscopic power morcellation, based on a preoperative diagnosis of leiomyoma, was LGESS, which led to abdominal re-exploration in many cases [8].

Given its high spatial resolution and excellent soft-tissue contrast, magnetic resonance imaging (MRI) has been extensively used to differentiate ESS, including LGESS, from other benign uterine tumors. As such, several studies have identified characteristic MRI features of ESS, including irregular margins, intramyometrial nodular extensions, intra-tumoral T2 low signal intensity (SI) bands, T2 low SI rims, and diffusion restrictions [3,9,10,11]. However, the interpretation of these features is limited by the rarity of ESS. To our knowledge, there are no previous case reports of LGESS that emphasize the unexpected T1 high signal intensity of T2 low SI rims or the diffusion restriction pattern associated with T2 low SI bands.

Herein, we present a case of LGESS in a 28-year-old female patient that clinically and radiologically mimicked red degeneration of a leiomyoma (RDL), with a special focus on specific MRI findings, including a peripheral rim showing T1 hyperintensity with T2 hypointensity and its pathologic correlation, as well as a diffusion restriction pattern in relation to previously reported studies.

## 2. Case Report

A 28-year-old nulliparous woman was referred to our institution for further evaluation and proper management after sonographic findings at a local clinic suggested a uterine leiomyoma with degeneration. She presented with irregular vaginal bleeding and sharp pelvic pain lasting for one week. The patient had no significant medical history, and gynecological examination revealed a markedly enlarged uterus. On ultrasound examination, a mixed echoic intramural mass including both hyperechoic and hypoechoic components was observed in the posterior wall of the uterus. Blood test results were within normal limits, with the CA-125 concentration measured at 12.9 IU/mL.

MRI performed for further characterization revealed a well-defined, 6.3 cm-sized, round mass in the posterior wall of the uterus, located within the myometrium, with an intact endometrial–myometrial border and without evidence of endometrial thickening. The lesion exhibited relatively homogeneous high SI on T2-weighted imaging (T2WI) and iso- to slightly low SI on T1-weighted imaging (T1WI) compared to the surrounding myometrium. A peripheral rim surrounding the mass appeared hyperintense on T1WI and hypointense on T2WI. Several intralesional areas showed linear low SI on both T1WI and T2WI. On contrast-enhanced T1WI, the mass showed no enhancement. On diffusion-weighted imaging (DWI), parts of the lesion exhibited linear high SI due to restricted diffusion, while most of the lesion showed low SI (Figure 1). No enlarged lymph nodes or fluid collection were identified in the pelvic cavity. The preoperative diagnosis was RDL.

Intraoperatively, her uterus appeared globular, with an intramural mass located in the posterior body near the fundus, initially suspected to be a uterine myoma with degeneration. The patient’s postoperative recovery was uneventful, and she was discharged in stable condition on postoperative day 4.

Upon histopathological examination of the excised specimen, a soft mass was observed macroscopically, with a cut surface that exhibited brownish degeneration. Microscopically, the specimen demonstrated diffuse necrosis. The tumor cells showed diffuse positivity for CD10 and focal positivity for actin but were negative for Ki67, desmin, and S100 (Figure 2). Based on these results, the tumor was diagnosed as an endometrial stromal tumor with extensive necrosis, likely representing a low-grade endometrial stromal sarcoma.

Although total abdominal hysterectomy with bilateral salpingo-oophorectomy was recommended as the primary treatment option, further evaluation to explore fertility-preservation treatment was also considered given the patient’s young age. However, she chose to forgo additional assessments and was subsequently lost to follow-up.

## 3. Discussion

While laparoscopic morcellation is widely used for benign uterine tumors, including leiomyomas, due to benefits like shorter postoperative hospital stays and reduced intraoperative and postoperative risks, it carries the risk of spreading undiagnosed malignant tissue beyond the uterus [12]. Therefore, when malignancy is suspected, a more invasive surgical approach, such as laparotomy without morcellation, is preferred to mitigate the risk of malignancy dissemination, despite its association with higher postoperative morbidity, longer recovery periods, and increased healthcare costs [13,14]. Additionally, intraoperative frozen section analysis is often inadequate for accurate differentiation due to limited tumor sampling, making it difficult to distinguish LGESS from leiomyoma or adenomyosis [15,16]. Thus, preoperative distinction and risk assessment using imaging findings are crucial for developing an appropriate treatment plan.

ESS often presents as a large uterine mass with low SI on T1WI and heterogeneous moderate to high SI on T2WI, accompanied by persistent heterogeneous enhancement following contrast administration on MRI [17]. Regarding the tumor location within the uterus, some studies suggest that ESS originates in the endometrium, growing as a polypoid endometrial tumor that invades the myometrium [18]. In contrast, other studies report a greater frequency of ESS in the myometrium, where it exhibits imaging features similar to leiomyomas with degenerative changes [3,11]. Irregular margins, nodular lesions at the periphery, and intramyometrial nodular extensions are more frequently associated with ESS [9]. Others have reported intra-tumoral low SI bands on T2WI in areas of myometrial invasion, often creating a “bag of worms” appearance [10]. High SI on DWI with low SI on apparent diffusion coefficient (ADC) mapping [3] may also help to characterize ESS. However, our case did not fully align with these features, presenting as an intramural mass with well-defined margins and lack of enhancement. Although subtle intra-tumoral low SI on T2WI and some areas of restricted diffusion were observed, the overall presentation differed. The aforementioned studies had small sample sizes, with fewer than 13 cases of LGESS, and their focus was on imaging findings of ESS cases as a group, including high-grade ESS cases, rather than focusing on LGESS.

A limited number of studies have examined the imaging findings of LGESS alone due to its rarity. To differentiate LGESS from leiomyoma, Furukawa et al. suggested that a hypointense rim on T2WI may be a characteristic feature of myometrial LGESS, corresponding to fibrous tissue or a mixture of tumor and smooth muscle cells [11]. Although a similar hypointense rim is often seen, RDL typically presents with a hyperintense rim on T1WI as well [19], whereas myometrial LGESS lacks this T1 hyperintense feature [11]. In our case, although a hypointense rim was noted on T2WI, the rim exhibited hyperintensity on T1WI, which contributed to the preoperative diagnosis of RDL. Additionally, a previously published case report of LGESS portrayed a T1 hyperintense/T2 hypointense rim around the mass in their figures, although the authors did not explicitly mention this as a T1 hyperintense feature [20]. This observation suggests that a T1 hyperintense rim could be present in other cases of LGESS, despite its lack of emphasis in prior reports.

To further characterize this rim, susceptibility-weighted imaging (SWI) would have been useful. Since the rim in RDL reflects thrombosed vessels filled with red blood cells, and SWI is highly sensitive to susceptibility effects and finely tuned to detect blood products, it typically shows a more prominent dark signal on SWI [21]. Although SWI was not performed, as it was not part of our routine pelvic MRI protocol for gynecologic evaluations, considering its limitations such as longer acquisition times and susceptibility to magnetic and motion artifacts [22], it could have provided valuable insights into the nature of the rim, potentially helping to avoid a preoperative misdiagnosis of RDL.

To our knowledge, a published multicenter study involving 25 cases of LGESS, compared to degenerated leiomyomas, is the largest imaging study on LGESS to date. Based on findings from a smaller series, those authors aimed to identify specific MRI features of LGESS, reporting the three significant T2WI findings of intra-tumoral low SI bands, cystic/necrotic changes, and the absence of a speckled appearance typical of leiomyomas. Although DWI showed high sensitivity, its ability to distinguish LGESS from rare leiomyoma variants was limited, as the two often exhibit similar high or isointense patterns. Instead, the normalized ADC value was significantly lower in LGESS compared to rare leiomyomas [6]. In our case, the three characteristic MRI features on T2WI were observed, though the intra-tumoral low SI bands were subtle. Additionally, some areas showed diffusion restriction. The normalized ADC value in our case was calculated as 0.207, which aligns with the study’s suggested optimal cutoff values for diagnosis of LGESS, defined as ≤0.31 and ≤0.27 by two readers. As two of the T2 features (intra-tumoral low SI bands and cystic/necrotic changes) can also appear in rare leiomyomas, potentially reducing specificity, the study suggested combining the absence of a speckled appearance on T2WI with qualitative and quantitative ADC evaluation to enhance diagnostic accuracy [6].

Regarding the pattern of diffusion restriction, no studies have focused on the pattern of diffusion restriction associated with T2 low SI bands. Although one study examined DWI with ESS, it primarily emphasized the ADC value with only brief mention of the site of diffusion restriction, noting that hyperintensity clearly delineated the tumor borders [3]. In our case, we observed high SI on DWI with corresponding low SI on the ADC along the T2 low SI bands, which did not exhibit diffusion restriction. Koyama et al. reported that these T2 low SI bands represent preserved myometrial fiber bundles separated by clusters of tumor cells several millimeters in size [10]. Their histopathologic correlation suggests that the observed diffusion restriction in our case indicates a viable tumor along myometrial infiltration sites. Notably, these diffusion-restricted areas showed no enhancement. Several studies support the possibility of viable tumor presence in areas with diffusion restriction but without enhancement [23,24], which might help to explain the imaging findings in our case.

As previously noted, imaging findings for LGESS vary significantly despite its rarity, complicating preoperative diagnosis. This variability may arise from differences in cellular origin. Some studies on the heterogeneity of ESS origins suggest that myometrial ESS may develop from undifferentiated mesenchymal elements, while endometrial ESS originates from stromal tissue within the endometrium [3,25]. These differences in cellular origin and differentiation contribute to the variability in presentation and imaging features among ESS cases. In a similar context, one study observed intra-tumoral T2WI low SI bands in tumors involving the myometrium but absent in those confined to the endometrium [10], suggesting that tumor location might influence imaging characteristics. Therefore, depending on whether LGESS originates in the endometrium or myometrium, imaging findings may differ, with potential greater diversity observed in myometrial LGESS. This can help to explain the variability in reported LGESS imaging features.

## 4. Conclusions

In this report, we present a case of LGESS with a peripheral rim exhibiting high SI on T1WI and low SI on T2WI, which led to a preoperative misdiagnosis of RDL. We hypothesize that SWI could have been valuable in this case given its high sensitivity to blood products and susceptibility effects, ultimately aiding in distinguishing RDL from LGESS. Moreover, DWI restriction was observed along the T2 low SI bands, which themselves showed no diffusion restriction. We propose that the viable tumor along the preserved myometrial fibers appears as a diffusion-restricted area adjacent to non–diffusion-restricted bands in our case.

Further research and case accumulation are essential to identify imaging features that differentiate LGESS from other uterine diseases. As demonstrated by the utility of ADC mapping, advanced MRI sequences such as SWI may play an increasingly important role in improving diagnostic accuracy for LGESS, especially when T2 low SI rims are observed.

## Figures and Tables

**Figure 1 diagnostics-15-00018-f001:**
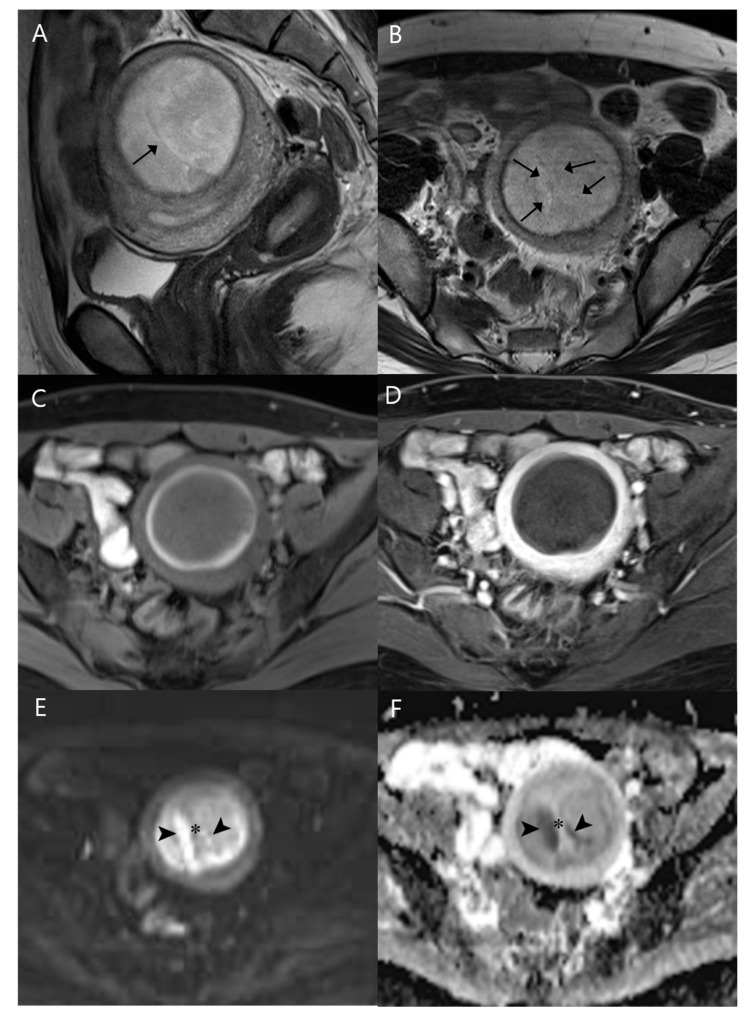
MRI of the tumor: (**A**) sagittal and (**B**) axial T2-weighted images show a 5.8 × 5.8 × 5.2 cm mass within the posterior myometrium, exhibiting well-defined margins with intra-tumoral low signal intensity bands (arrows) and a peripheral low signal intensity rim. (**C**) On axial fat-suppressed T1-weighted imaging, the tumor shows iso- to slightly low signal intensity with a peripheral high-signal intensity rim. (**D**) On axial contrast-enhanced T1-weighted imaging, the tumor shows no enhancement (**E**). Diffusion-weighted image and (**F**) a corresponding apparent diffusion coefficient map show certain areas (arrowheads) of the lesion with high signal intensity and low apparent diffusion coefficient values along the T2 low signal intensity band (asterisk), which itself shows no diffusion restriction.

**Figure 2 diagnostics-15-00018-f002:**
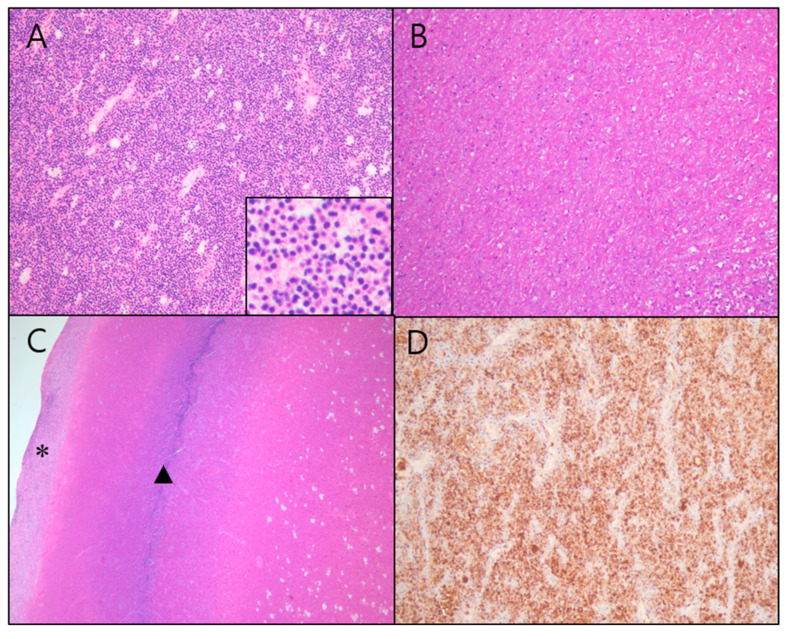
Histologic features of the tumor: (**A**) the tumor is densely cellular, showing uniform small-sized tumor cells with scant cytoplasm, round to oval nuclei, and inconspicuous nucleoli (H&E stain, original magnification: ×200) (inset: H&E stain, original magnification: ×400). (**B**) A large portion of the tumor shows necrosis (H&E stain, original magnification: ×200). (**C**) Histological image of the peripheral region of the tumor, corresponding to the T1 hyperintense and T2 hypointense rims on MRI, shows necrotic tissue (triangle), with normal uterine tissue visible beyond the outer margin (asterisk). (H&E stain, original magnification: ×200). (**D**) The tumor cells are positive for CD10 (original magnification: ×200).

## Data Availability

No new data were created or analyzed in this study. Data sharing is not applicable to this article.

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
