# Peer review of "Low-Grade Endometrial Stromal Sarcoma: A Case Report of a Rare Uterine Malignancy Mimicking Degenerative Uterine Leiomyoma in a Nulliparous Woman"

_diagnostics, 2024, doi:10.3390/diagnostics15010018_

Round 1

Reviewer 1 Report

Comments and Suggestions for Authors

I would like to thank the authors for their case presentations of a rare disease, low-grade endometrial stromal sarcoma.

Although it is rare, I could not understand what additional information it provides to the articles I have mentioned below.

https://doi.org/10.1093/bjrcr/uaad012

doi: 10.1038/s41598-021-98473-z

In the introduction of the study, ESS and definitions, epidemiological data are well presented, the reason for presenting the case report and its importance are not well explained.

I would like to mention whether there will be differences in treatment algorithms in cases where the diagnosis is Degenerative Uterine Leiomyoma or low-grade ESS.

Kind regards

Author Response

Response to Reviewer 1 Comments

I would like to thank the authors for their case presentations of a rare disease, low-grade endometrial stromal sarcoma.

-> Thank you for your valuable feedback and for taking the time to review our case presentation of low-grade endometrial stromal sarcoma (ESS).

R1-1:  Although it is rare, I could not understand what additional information it provides to the articles I have mentioned below.

https://doi.org/10.1093/bjrcr/uaad012

doi: 10.1038/s41598-021-98473-z

-> We sincerely appreciate your insightful comments and for referencing the case report, which was similarly misinterpreted as a degenerated myoma, as well as another relevant article that is included in our reference list. We agree that our case shares similarities with the referenced case, particularly the intratumoral low signal intensity (SI) bands and peripheral low SI rim on T2WI. However, we would like to highlight the differences in the suggested types of degeneration on MR imaging, specifically cystic degeneration versus red degeneration (RDL), and the T1 signal intensity of the peripheral rim.

While a T1 hyperintense/T2 hypointense rim is recognized as suggestive of RDL, a T1 non-hyperintense/T2 low SI rim has been reported in LGESS cases. The T1 signal intensity of the rim is therefore a crucial differential point between RDL and LGESS (doi: 10.1002/jmri.22126). Our case, however, presented a T1 hyperintense rim, leading to an incorrect diagnosis of RDL. We hope that this distinction will help clarify the differences between the cases.

To emphasize this distinction, we have rephrased and added a more detailed description in the Introduction section (Page 2, Introduction, line 85) (Page 6, Discussion, line 188).

Page 2, Introduction, line 85

 Herein, we present a case of LGESS in a 28-year-old female patient that clinically and radiologically mimicked red degeneration of a leiomyoma, with a special focus on specific MRI findings, including a peripheral rim showing T1 hyperintensity with T2 hypointensity and its pathologic correlation, as well as a diffusion restriction pattern in relation to previously reported studies.

Page 6, Discussion, line 188

In our case, although a hypointense rim was noted on T2WI, the rim exhibited hyperintensity on T1WI, which contributed to the preoperative diagnosis of RDL.

R1-2: In the introduction of the study, ESS and definitions, epidemiological data are well presented, the reason for presenting the case report and its importance are not well explained.

-> Thank you for your valuable comment. To highlight the reasoning and its significance, as suggested by your feedback, we have made modifications and additions in the Introduction section (Page 2, Introduction, line 75), (Page 2, Introduction, line 81)

Page 2, Introduction, line 75

As such, several studies have identified characteristic MRI features of ESS, including irregular margins, intramyometrial nodular extensions, intra-tumoral T2 low-signal-intensity (SI) bands, T2 low-SI rims, and diffusion restrictions. However, the interpretation of these features is limited by the rarity of ESS. To our knowledge, there are no previous case reports of LGESS that emphasize the unexpected T1 high signal intensity of T2 low-SI rims or the diffusion restriction pattern associated with T2 low-SI bands.

Page 2, Introduction, line 85

Herein, we present a case of LGESS in a 28-year-old female patient that clinically and radiologically mimicked red degeneration of a leiomyoma, with a special focus on specific MRI findings, including a peripheral rim showing T1 hyperintensity with T2 hypointensity and its pathologic correlation, as well as a diffusion restriction pattern in relation to previously reported studies.

R1-3: I would like to mention whether there will be differences in treatment algorithms in cases where the diagnosis is Degenerative Uterine Leiomyoma or low-grade ESS.

-> We appreciate your insightful comment. As you pointed out, we have added further explanation regarding the differences in treatment algorithms for cases diagnosed as Degenerative Uterine Leiomyoma versus low-grade ESS in the Discussion section (Page 5, Discussion, line 149)  

Page 5, Discussion, line 149

 While laparoscopic morcellation is widely used for benign uterine tumors, including leiomyomas, due to benefits like shorter postoperative hospital stays and reduced intraoperative and postoperative risks, it carries the risk of spreading undiagnosed malignant tissue beyond the uterus While laparoscopic morcellation is widely used for benign uterine tumors due to benefits, such as shorter postoperative hospital stays and lower intraoperative and postoperative risks, it carries the risk of spreading undiagnosed malignant tissue beyond the uterus [12]. Therefore, when malignancy is suspected, a more invasive surgical approach, such as laparotomy without morcellation, is preferred to mitigate the risk of malignancy dissemination, despite its association with higher postoperative morbidity, longer recovery periods, and increased healthcare costs [13,14].

Reviewer 2 Report

Comments and Suggestions for Authors

In this case report the authors present a case of LGESS with a peripheral rim exhibiting high SI on T1WI and low SI on T2WI, which was wrongly diagnosed as RDL. The case report is well written, and the results are clearly presented.

I would like the authors to better discuss the costs (logistic/economic/health-related for the patient) and benefits associated with a more invasive surgical approach when malignancy is suspected (discussion -line 142).

Why susceptibility-weighted imaging was not performed if the authors themselves state that would have been useful in this case? Which are the limitations associated with this sequence?

Author Response

Response to Reviewer 2 Comments

General comment: In this case report the authors present a case of LGESS with a peripheral rim exhibiting high SI on T1WI and low SI on T2WI, which was wrongly diagnosed as RDL. The case report is well written, and the results are clearly presented.

-> Thank you very much for taking the time to review our paper and for your kind feedback. We sincerely appreciate your positive comments on our case report. We are glad that the case was clearly presented and the results well-received. Your acknowledgment of our work is truly encouraging, and we hope our report contributes meaningfully to the understanding and diagnosis of LGESS.

R2-1: I would like the authors to better discuss the costs (logistic/economic/health-related for the patient) and benefits associated with a more invasive surgical approach when malignancy is suspected (discussion -line 142).

-> We sincerely appreciate your valuable advice. We completely agree that a better discussion of more invasive surgical approaches is needed. To address this, we have added and modified the Discussion section (Page 5, Discussion, line 155).

Page 5, Discussion, line 155

Therefore, when malignancy is suspected, a more invasive surgical approach, such as laparotomy without morcellation, is preferred to mitigate the risk of malignancy dissemination, despite its association with higher postoperative morbidity, longer recovery periods, and increased healthcare costs [13,14].

R2-2: Why susceptibility-weighted imaging was not performed if the authors themselves state that would have been useful in this case? Which are the limitations associated with this sequence?

-> We apologize for our unclear statement on this in the original manuscript. As you commented, we have rephrased the text and added to clarify this point in the Discussion (Page 6, Discussion, line 200)

Page 6, Discussion, line 200

Although SWI was not performed, as it was not part of our routine pelvic MRI protocol for gynecologic evaluations, considering its limitations, such as longer acquisition times and susceptibility to magnetic and motion artifacts[1], it could have provided valuable insights into the nature of the rim, potentially helping to avoid a preoperative misdiagnosis of RDL

Round 2

Reviewer 1 Report

Comments and Suggestions for Authors

I would like to thank the authors for their positive feedback and revisions to my critiques. I think that it will make a limited contribution to the literature in its current form, but I think it can be accepted.

Best regards